# Healthcare professionals' awareness of the levels of disciplinary collaboration in older adults' care during clinical practice in Nigeria: A multi-method study

Chukwunonso Geoffrey Okafor[1⊙], Augustine Chukwuebuka Okoh[2,3⊙]*, Adaobi Odega[1,2‡], Stanley Maduagwu[1‡], Michael Kalu[2,4⊙]

1 Department of Medical Rehabilitation, Faculty of Health Sciences and Technology, College of Health Sciences, Nnamdi Azikiwe University, Nnewi Campus, Nnewi, Nigeria, 2 Emerging Researchers & Professionals in Ageing – African Network, Nigeria (www.erpaan.org), Port Harcourt, Nigeria, 3 Department of Health Research Methods, Evidence and Impact, McMaster University, Hamilton, Ontario, Canada, 4 School of Kinesiology and Health Science, Faculty of Health, York University, Ontario, Canada

⊙ These authors contributed equally to this work.
‡ These authors also contributed equally to this work.
* okoha@mcmaster.ca

## Abstract

### Introduction

Collaborative care has been recognized as a fundamental aspect of geriatric/gerontological care, as it can enhance health and social outcomes among older adults. However, very little is known about Nigerian healthcare professionals' (HCPs) awareness of disciplinary collaborations in older adults' care. This study investigated the HCPs' awareness (level of awareness, definitions, and perceived practice) of the disciplinary collaborations in older adults' care during clinical practice at a tertiary hospital in the South-Eastern region of Nigeria.

### Methods

We conducted a multi-method study involving 245 HCPs who responded to authors' developed questionnaire. Data was analyzed using descriptive and inferential statistics.

### Results

The level of awareness of disciplinary collaboration varies: intradisciplinary (65.7%), multidisciplinary (72.1%), cross-disciplinary (57.1%), interdisciplinary (76.3%), and transdisciplinary (40.8%) collaborations. Having a Bachelor's degree (OR = 1.055, 95% CI [−0.154, 5.662], $p = 0.038$) and a transdisciplinary education (OR = 1.023, 95% CI [0.904, 6.646], $p = 0.010$) predicted better awareness of the inter-/trans] disciplinary collaborations. There was a significant change in the participants' level of

**Data availability statement:** Our data set is publicly available from the Zenodo database (https://zenodo.org/records/15238097).

**Funding:** The author(s) received no specific funding for this work.

**Competing interests:** NO authors have competing interests.

awareness of the type of disciplinary practice in their hospital before and after seeing the description of the collaborative practice constructs [$X^2$ (2)=29.747, $p < 0.001$]. Barriers to higher collaborative practices in geriatric/gerontological care were inequitable remuneration ($X^2$(2) = 4.332, $p = 0.037$), lack of integrated electronic medical records ($X^2$(2) = 12.562, $p = 0.002$), and a limited number of specialists ($X^2$(2) = 54.093, $p < 0.001$).

## Conclusion

Achieving consensus on the definition of intra-, cross-, multi-, inter-, and transdisciplinary practice is essential for a clear understanding of the constructs, facilitating its proper implementation in geriatric care and the development of effective solutions to systemic barriers. A transdisciplinary educational curriculum could support the uptake of transdisciplinary collaborative geriatric practice models in Nigeria.

## Introduction

Globally, the population of older adults is increasing rapidly, and by 2050, individuals aged 60 years and over are projected to outnumber children aged 0–14 [1]. In 2019, Sub-Saharan Africa (SSA) had 31.9 million older adults (65 + years), a figure that is expected to rise to 101.4 million by 2050, reflecting a 218% increase and ranking as the second-highest percentage change globally after Northern Africa and Western Asia [2]. As the most populous country in SSA, Nigeria is projected to contribute significantly to this demographic shift, with its proportion of older adults expected to surpass that of other countries in the region [3]. These projections underscore the urgent need to plan for population aging by exploring healthcare disciplinary practice models and practices in geriatric/gerontological care within the Nigerian context.

Older adults are likely to have multiple chronic health conditions that affect their physical, social, and cognitive functioning [4]. Interprofessional efforts are often required to manage the complex range of chronic conditions with which older adults present [3]. The American Geriatric Society promotes the adoption of collaborative approaches to geriatric practice [5]. Several studies have shown that compared to the effort of a single discipline, effective teamwork among professionals from different healthcare disciplines reduces adverse events, mortality, and the length of hospital stay, while also improving quality of life and satisfaction with service provision [6–8].

The conceptualization of collaborative practice and the terminology used to describe the levels of collaboration are often used inconsistently in the literature [3]. To clarify them, Jensenius [9] described five different disciplinary practice levels in hierarchical order representing increasing levels of partnership, information sharing, and coordination in healthcare teams: intradisciplinary, multidisciplinary, cross-disciplinary, interdisciplinary, and transdisciplinary (See Fig 1).

Intradisciplinary collaboration refers to collaboration among professionals within the same discipline or field of expertise—for instance, between orthopedic and geriatric physiotherapists [3]. The other terms – multidisciplinary, cross-disciplinary,

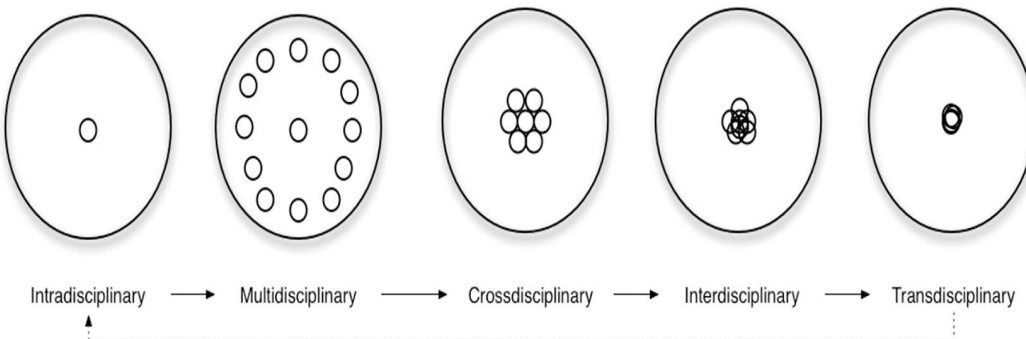

**Fig 1. Pictorial illustration and written description of disciplinary practice models.** Sources: Choi & Pak [11]; Jensenius [9].

**INTRADISCIPLINARY**

- Patients are seen by a single discipline

**MULTIDISCIPLINARY:**

- Patients' care and management are done by different healthcare professionals (HCPs), with each discipline approaching the patients from their perspective.
- Each discipline individually consults with the patient, i.e. patients can move from one department to another (e.g. from orthopedic unit to physiotherapy unit).

**CROSSDISCIPLINARY:**

- Patients are still individually consulted by different disciplines (professionals) but have started having regular case meetings to discuss the patients' condition and progress.
- Professionals with close disciplinary boundaries often consult together (e.g. occupational therapy and physiotherapy working together).

**INTERDISCIPLINARY:**

- Patients' cases, including short- and long-term management goals, are reviewed by the healthcare professional team together with the patient, often one time during ward rounds.
- Even though patients' cases are reviewed by all HCPs at one time, there is still disciplinary boundary and recognition.

**TRANSDISCIPLINARY:**

- Patients' history, assessments, diagnosis, and short- and long-term management goals are discussed with the patients/relatives at a single consultation by all healthcare professional teams. A designated health assistant is employed to take notes in one care plan template (often computerized), where all HCPs can have access. This single consultation often starts at the beginning of the patient's care (on admission or at accident and emergency).
- All HCPs are treated equally.
- The patient is empowered in the decision-making of his or her care.

In transdisciplinary care, a case manager for a patient is appointed, depending on the area of care that is most important to the patient. *Example:* If a patient's goal is to lose weight through surgery, a gastroenterologist and plastic surgeon might be the case managers.

interdisciplinary and transdisciplinary entail collaborations across different disciplines or fields. They, however, differ in the level of integration and interactions among professionals from different disciplines. In a multidisciplinary approach, the professionals maintain their disciplinary boundary and contribute their specific knowledge to address a common problem [10,11]. Cross-disciplinary entails actively seeking insights and integrating perspectives from diverse fields outside their disciplinary boundaries, going beyond individual disciplines to bridge disciplinary gaps and generate innovative solutions [9]. An interdisciplinary approach entails greater interactions, focusing on synthesizing and integrating knowledge across

disciplines to create a shared care goal [12]. A transdisciplinary approach to care entails more collaboration and integration to generate solutions that emerge from the blending and transcending disciplinary perspectives [12]. In transdisciplinary care, individual disciplinary perspectives are blurred, leading to more shared decision-making, problem-solving, and co-creation techniques.

Disciplinary collaboration in geriatric practice has not been fully explored among healthcare professions in Nigeria [3]. Currently, few empirical studies have explored aspects of disciplinary collaboration among HCPs in Nigeria [3,8,13]. While Falana and colleagues' [13] study explored disciplinary collaboration between physicians and nurses practicing in a public hospital in South-West Nigeria, Iyoke et al. [8] explored physician-physician collaboration in an obstetric department of another public hospital in South-East Nigeria. These two studies, Falana et al. [13] and Iyoke et al. [8], provided insights into collaborative practices in Nigeria, specifically focusing on intra-disciplinary and multidisciplinary approaches respectively. However, while they are underreasearched in this region, available evidence has shown that interdisciplinary and transdisciplinary care approaches produce better geriatric care outcomes compared to others [14].

To further explore this, our team conducted a qualitative case study in a geriatric unit, investigating the views of Nigerian healthcare workers on transdisciplinary approaches to older adults' care. Our findings identified five key features of transdisciplinary practice—consolidated consultation, consolidated care documentation, care file accessibility, shared care decision-making, and a designated care manager. We also observed underlying issues associated with these constructs. Summarily, disciplinary collaboration (of any type – inter- or multidisciplinary) among HCPs is still evolving as a practice model in the Nigerian context, partly due to the historic conflict between HCPs prompted by (a) frictions arising from doctors' dominance in the Nigerian healthcare system; (b) the lack of understanding of the roles each healthcare worker plays in the care of older adults; and (c) HCPs' training often being siloed – the students are trained with little or no communication with each other [3]. Therefore, considering these issues and recommendations from a previous study [3], it is crucial to conduct a quantitative investigation to explore the awareness of disciplinary practice in another Nigerian hospital that has recently established a geriatric unit.

Uncovering the range of collaborative practices available in this context may stimulate dialogue on strategies and interventions to foster more cohesive teamwork among the HCPs. Examining this issue would provide valuable insight into HCPs' awareness of disciplinary collaboration in the care of older adults. The study findings may also highlight the factors (facilitators and barriers) that influence HCPs' views on disciplinary collaboration. It would illuminate areas of underperformance that require improvements as well as areas of progress that can be further strengthened. Thus, this study sought to investigate the HCPs' awareness of the disciplinary collaborations in older adults' care during clinical practice in a tertiary hospital in the South-Eastern region of Nigeria. Specifically, this study sought to (1) describe the participants' level of awareness of the disciplinary collaboration constructs and barriers to effective collaborative practice in the study site; (2) ascertain the influence of demographic variables (e.g., sex, educational qualification) and practice variables (e.g., profession, specialty, years of practice) on participants' level of awareness of the disciplinary collaboration constructs; and (3) determine the changes in participants' awareness of the collaborative practice in the study site after reviewing the description of the disciplinary constructs.

## Methods

### Study design

This study employed a multi-method approach comprising both cross-sectional study design (Aims 1 and 2) and pretest-posttest study design (Aim 3). Data were collected between January 3, 2022 and April 30, 2022 using both a self-administered paper-based (face-to-face) and a web-based version of the questionnaire. We obtained ethical approval from the University of Southampton Ethics and Research Governance Online (ERGO II 48589) and approval from the NAUTH (local) Institutional Review Board (NAU/FHST/2021/MRH48). Participants read the study information sheet and

completed a written consent form before filling out the questionnaire. We followed Strengthening the Reporting of Observational Studies in Epidemiology (STROBE) checklist for cross-sectional studies [15].

## Study setting

The study was conducted at Nnamdi Azikiwe University Teaching Hospital (NAUTH), in Nnewi, Anambra state, South-Eastern region of Nigeria. NAUTH is a tertiary hospital which has several health and social care professionals. The hospital provides a broad range of services, which include but not limited to medical, surgical, diagnostic, rehabilitative, dietetic, social support services, etc. It has a wide coverage, with a catchment population of almost 31 million people. Most health and social care professionals are full time staff [16].

## Sampling and recruitment

We recruited participants through multiple methods, including word of mouth, distribution of recruitment flyers in strategic locations within the hospital, and targeted professional WhatsApp groups. Using a convenience sampling technique, healthcare workers were recruited if they (a) were full-time staff of NAUTH; (b) have at least two years of experience of providing care to older adults; (c) self-identified as a licensed professional in their respective profession at the time of this study, as no records were publicly available to verify their licensure; and (d) were willing to participate in the study. The sample size (n) was calculated using the sample size determination formula for an unknown population as described by Naing et al. [17].

$$n = \frac{Z^2 \, P(1 - P)}{d^2}$$

We assumed a Z statistic for a level of confidence (Z) of 95% (95% CI corresponds to Z value of 1.96), a degree of precision (d) of 5%, and expected prevalence or proportion (P) as 80%. Thus, the study's estimated sample size was 245.

## Questionnaire structure

The questionnaire is an anonymized, semi-structured questionnaire composed of 35 questions distributed across six sections. Sections 1 and 2 consist of demographic and practice-related information (n = 12); sections 3 and 4 consist of questions on awareness of the levels of disciplinary collaborations in older adults' care (n = 15); and sections 5 and 6, respectively, consists of questions on barriers to effectively practice higher levels of (i.e., inter- and trans-) disciplinary collaborations (n = 8) in their practice setting. Of note, in section 4, participants rated the level of disciplinary collaboration in their practice setting after reviewing pictorial depiction of the disciplinary levels and repeated their rating after reviewing written description of the disciplinary levels. We described the questionnaire's face and content validation process in the supporting information in S1 File.

## Data collection

Individuals who met the inclusion criteria described above were given the Participant Information Sheet and Consent Form; and the survey questionnaires (research instrument) with assigned identifiers. The questionnaire was administered with two methods. A sealed envelope containing the questionnaire, Participant Information Sheet and Consent Form, and a return postal stamp was mailed (postal) to the participants that preferred the paper format. Then, Google Forms©, a secured online survey service, containing the questionnaire, Participant Information Sheet and Consent Form was e-mailed to the others. In order to increase participation in the study, we utilized Dillman's Tailored Design methods [18], employing the following approaches: personalized email invitations were sent to potential participants, highlighting the importance of their involvement, and emphasizing the significance of their responses in achieving meaningful results.

Weekly personalized email reminders were sent to participants throughout the study, with a higher frequency of daily reminders during the final three days before the questionnaire's closure.

## Data analysis

Data were analyzed using SPSS (IBM SPSS Statistics for Windows, Version 25.0, 2017, SPSS, Armonk, NY). Descriptive statistics of mean, standard deviation, frequencies, and percentages were used in the analysis of participants' demographic variables. All the data were categorical variables. To answer study aim 1, we employed descriptive statistics, crosstabulation, and Chi-square testing to ascertain the factors associated with high levels of disciplinary collaboration on the study site. To answer study aim 2, a binomial logistic regression model was used to predict awareness of the disciplinary collaboration constructs (inter-/transdisciplinary), owing to a binary dependent variable (awareness [yes or no]). To answer study aim 3, we performed a Friedman test, followed by a Wilcoxon signed-rank test for post hoc, to test the influence of pictorial illustrations and written descriptions of the disciplinary practice constructs on participants' awareness of the five levels of disciplinary collaboration practiced in the study setting. Using the Friedman test, we analyzed the sums of ranks across pre- and post-test to determine if there was any significant difference in participants' awareness level. The post hoc (Wilcoxon signed-rank test) examined the intervention (pictorial illustration or written description) that may led to the significant difference in awareness level. The constructs were treated as ordinal variables ranging from intradisciplinary (lowest) to transdisciplinary (highest). The level of significance was set at $\alpha = 0.05$.

## Results

### Participants' socio-demographic variables

Out of the 324 healthcare professionals (HCPs) contacted to participate in the study, 256 HCPs completed the questionnaire, resulting in a 79% response rate. However, due to 11 incomplete questionnaires, only 245 completed questionnaires were included in the analysis. The participants were between 22 and 65 years old (mean age of $34.9 \pm 7.6$) and included more males (58.4%) than females. As shown in Table 1, a large proportion of the participants had at least a Bachelor of Science degree (82.0%), were nurses (35.9%), and specialized in orthopedics (31.8%). Also, 75.1% of the respondents had a basic certificate in geriatrics (not graduate degree in geriatrics), 60.4% provided care for 1–5 elderly patients weekly, and 49.8% had 2–5 years of experience in the care of older adults.

### Participants' awareness of the levels of disciplinary collaboration

Participant awareness was highest for interdisciplinary collaboration (76.3%), followed by multidisciplinary collaboration (72.1%) and intradisciplinary collaboration (65.7%). In contrast, lower levels of awareness were reported for cross-disciplinary collaboration (57.1%) and transdisciplinary collaboration (40.8%).

### Barriers to inter-/transdisciplinary collaboration in the study setting

Regarding the barriers to high levels of disciplinary collaboration in the study site, we found significant association with inadequate specialists or HCPs with geriatrics certification ($X^2(2) = 54.093$, $p < 0.001$), lack of integrated electronic medical records ($X^2(2) = 12.562$, $p = 0.002$), and inequitable remuneration formula ($X^2(2) = 4.332$, $p = 0.037$). Specifically, 17.55%, 37.14%, and 31.84% of the respondents were sure that not having adequate specialists, lack of integrated electronic medical records (EMRs), and inequitable remuneration formula respectively were barriers to high level disciplinary practice in their hospital, as shown in Table 2.

### Regression model for predicting level of awareness among the participants

We performed a logistic regression model to determine the predictors of awareness of the disciplinary collaboration constructs among the participants in the study setting. The results in Table 3 shows that the logistic regression model was

**Table 1. Distribution of the participants' sociodemographic characteristics (N = 245).**

| Variable | Category | n (%) |
|---|---|---|
| Sex | Male | 143 (58.4) |
| | Female | 102 (41.6) |
| Highest Educational Qualification | Basic certificate | 10 (4.1) |
| | National diploma | 16 (6.5) |
| | Higher national diploma | 6 (2.4) |
| | Bachelor of Science | 201 (82.0) |
| | Master of Science | 12 (4.9) |
| Profession | Physician | 56 (22.9) |
| | Physiotherapist | 38 (15.5) |
| | Speech and Language pathologist | 9 (3.7) |
| | Orthopedic surgeon | 9 (3.7) |
| | Nurse | 88 (35.9) |
| | Medical Laboratory Science | 10 (4.1) |
| | Medical Radiographer | 35 (14.3) |
| Area of Specialty | Preoperative | 21 (8.6) |
| | Exercise and Sports Medicine | 6 (2.4) |
| | Radiology | 42 (17.1) |
| | Women's Health | 4 (1.6) |
| | Cardiopulmonary | 45 (18.4) |
| | Orthopedics | 78 (31.8) |
| | General Nursing | 15 (6.1) |
| | Generalist | 10 4.1) |
| | Medical Microbiology | 5 (2.0) |
| | Neurology | 19 (7.8) |
| Years of experience in geriatric practice | ≤ 1 year | 98 (40.0) |
| | 2-5 years | 122 (49.8) |
| | 6-10 years | 9 (3.7) |
| | 11-15 years | 16 (6.5) |
| Geriatric certification | Yes | 184 (75.1) |
| | No | 61 (24.9) |
| Number of older adults seen per week | 1-5 patients | 148 (60.4) |
| | 6-10 patients | 45 (18.4) |
| | 11-15 patients | 41 (16.7) |
| | > 15 patients | 11 (4.5) |

statistically significant ($X^2$ = 117.111; $p < 0.001$) suggesting that the model was a good fit. The model explained 50.8% of the variance in awareness level. Participants with a Bachelor of Science degree had a 5.5% greater likelihood of having a higher level of awareness of the disciplinary collaboration constructs than those with a basic certificate (OR = 1.055, 95% CI [−0.154, 5.662], $p = 0.038$). Exposure to a transdisciplinary education module during entry level training increases the odds of having a high level of awareness of the disciplinary collaboration constructs (OR = 1.023, 95% CI [0.904, 6.646], $p = 0.010$). Being a Medical Radiographer (OR = −0.014, 95% CI [−7.632, −0.952], $p = 0.012$) and specializing in Exercise and Sports Medicine (OR = −0.081, 95% CI [−4.718, −0.316], $p = 0.025$) decreases the odds of having a high level of awareness of the disciplinary collaboration constructs.

**Table 2. Barriers to achieving high level (inter- and trans-) of disciplinary collaboration in the study setting (N = 245).**

| Factors | Category | n (%) | $X^2$ | df | P |
|---|---|---|---|---|---|
| Poor communication | No | 0 (0.0) | 3.464 | 1 | 0.063 |
| | Maybe | 67 (27.35) | | | |
| | yes | 57 (23.27) | | | |
| Inadequate specialists | No | 59 (24.08) | *54.093 | 2 | <0.001 |
| | Maybe | 22 (8.98) | | | |
| | yes | 43 (17.55) | | | |
| Inadequate hospital supplies/equipment | No | 0 (0.0) | 0.016 | 1 | 0.900 |
| | Maybe | 37 (15.10) | | | |
| | yes | 87 (35.51) | | | |
| Lack of integrated EMRs | No | 0 (0.0) | *12.562 | 2 | 0.002 |
| | Maybe | 33 (13.47) | | | |
| | yes | 91 (37.14) | | | |
| Bill payment system | No | 7 (2.86) | 5.583 | 2 | 0.061 |
| | Maybe | 45 (18.36) | | | |
| | yes | 72 (29.39) | | | |
| Poor knowledge of the roles of other HCPs | No | 7 (2.86) | 1.878 | 2 | 0.391 |
| | Maybe | 48 (19.59) | | | |
| | yes | 69 (28.16) | | | |
| Interprofessional conflict | No | 0 (0.0) | 1.355 | 1 | 0.244 |
| | Maybe | 25 (10.20) | | | |
| | yes | 99 (40.41) | | | |
| Inequitable remuneration (wage) formula across the disciplines | No | 0 (0.0) | *4.332 | 1 | 0.037 |
| | Maybe | 46 (18.78) | | | |
| | yes | 78 (31.84) | | | |

*$p \leq 0.05$; EMR: Electronic Medical Record; HCPs: Healthcare professionals.

### Participants' awareness of the level of disciplinary collaboration at the study setting

There was a statistically significant difference in awareness of the level of disciplinary collaboration that is obtainable in the study site, $X^2(2) = 29.747$, $p < 0.001$. Post hoc analysis with Wilcoxon signed-rank tests was conducted with a Bonferroni correction applied, resulting in a significance level set at $p < 0.050$. The pictorial illustration (Fig 1) of the levels of disciplinary collaboration appears to have no significant influence in their awareness of disciplinary collaboration ($Z = -1.497$, $p = 0.134$). However, a significant difference in their awareness level was found after reviewing the written description ($Z = -5.351$, $p < 0.001$), as shown in Table 4. Their awareness of the level of disciplinary collaboration in their practice setting changed after seeing the meaning of the disciplinary practice constructs. At baseline, more respondents (49.8%) thought they practiced cross-disciplinarity but much fewer (2.0%) believed that they engaged in cross-disciplinary practice after seeing the meaning of those terminologies. Also, those who choose multidisciplinary practice increased from 20.4% at baseline to 61.6% after reading the written description.

## Discussion

This study examined healthcare professionals' awareness of the current state of disciplinary collaboration at tertiary hospital in the South-Eastern region of Nigeria and the current barriers to effective collaborative practice in their geriatric clinics. The study participants seem to be mostly aware of multidisciplinary and interdisciplinary practice. Majority of the

**Table 3. Predictors of high level of awareness of disciplinary collaboration among the Participants(N = 245).**

| Variable | Category | B | Exp(B) | p-value | 95% CI |
|---|---|---|---|---|---|
| Highest Educational Qualification | Basic certificate (ref) | | | | |
| | National diploma | −2.549 | 0.078 | 0.391 | −8.380, 3.282 |
| | Higher national diploma | −1.388 | 0.025 | 0.382 | −4.502, 1.726 |
| | Bachelor of Science | 2.908* | 1.055 | 0.038 | −0.154, 5.662 |
| | Master of Science | 0.347 | 1.415 | 0.676 | −1.280, 1.974 |
| Profession | Physician (ref) | | | | |
| | Physiotherapist | −1.209 | 0.298 | 0.089 | −2.605, 0.187 |
| | Speech and Language pathologist | −1.368 | 0.255 | 0.073 | −2.865, 0.129 |
| | Orthopedic surgeon | −2.264 | 0.104 | 0.152 | −5.336, 0.831 |
| | Nurse | 2.128 | 1.746 | 0.999 | −0.365, 4.621 |
| | Medical Laboratory Science | −0.651 | 0.521 | 0.409 | −2.197, 0.895 |
| | Medical Radiographer | −4.292* | 0.014 | 0.012 | −7.632, −0.952 |
| Area of Specialty | Preoperative (ref) | | | | |
| | Exercise and Sports Medicine | −2.517* | 0.081 | 0.025 | −4.718, −0.316 |
| | Radiology | 1.771 | 4.904 | 0.999 | −1.105, 4.647 |
| | Women's Health | 0.251 | 1.285 | 0.819 | −1.897, 2.399 |
| | Cardiopulmonary | 2.039 | 7.203 | 0.999 | −1.886, 5.964 |
| | Orthopedics | −0.495 | 0.610 | 0.598 | −2.335, 1.345 |
| | General Nursing | −1.274 | 0.280 | 0.117 | −2.869, 0.321 |
| | Generalist | −3.335 | 0.036 | 0.076 | −7.024, 0.354 |
| | Medical Microbiology | −0.783 | 0.457 | 0.792 | −6.600, 5.034 |
| | Neurology | −1.839 | 0.159 | 0.221 | −4.783, 1.105 |
| Years in experience in geriatric practice | ≤ 1 year (ref) | | | | |
| | 2-5 years | 0.662 | 1.939 | 0.699 | −2.699, 4.023 |
| | 6-10 years | 0.263 | 1.301 | 0.882 | −3.218, 3.744 |
| | 11-15 years | −2.179 | 0.113 | 0.326 | −6.524, 2.166 |
| Geriatric certification | No (ref) | | | | |
| | Yes | −0.497 | 0.608 | 0.386 | −1.622, 0.628 |
| Transdisciplinary education during entry level training | No (ref) | | | | |
| | Yes | 3.775** | 1.023 | 0.010 | 0.904, 6.646 |
| Constant | | 4.714 | 111.523 | | |
| Model $X^2$ (29) | | 117.111*** | | | |
| NagelKerke $R^2$ | | 0.508 | | | |

Note: *$p \leq 0.05$; **$p \leq 0.01$; ***$p \leq 0.001$.

participants believed that multidisciplinary collaboration is the most prevalent collaborative model in the study setting. However, participants' perceived practice level changed significantly after reviewing the written description of collaborative practice levels. The changes highlighted that participants thought their practice level was interdisciplinary and upon reviewing written description, they realized that it was a multidisciplinary approach to the care of older adults. This highlighted the need to provide consensus-contextual definitions of each construct to enhance HCPs' understanding and foster collaborative practices that are culturally-adapted to the Nigerian healthcare context. Moreso, it is possible that placing detailed descriptions of these disciplinary collaboration constructs in clinics would be a starting point to enhancing HCPs' understanding of these constructs.

**Table 4. Participants' awareness of the level of disciplinary collaboration in the study setting (N = 245).**

| X² | df | p | Post hoc | Z | P | Baseline awareness | n (%) | Post pictorial illustration | n (%) | Post written description | n (%) |
|---|---|---|---|---|---|---|---|---|---|---|---|
| 29.747 | 2 | < 0.001 | Pre-description – Post pictorial illustration | −1.497 | 0.134 | Intradisciplinary | 72 (29.4) | Intradisciplinary | 37 (15.1) | Intradisciplinary | 6 (2.4) |
| | | | Pre-description – Post written description | −5.351 | < 0.001 | Multidisciplinary | 50 (20.4) | Multidisciplinary | 169 (69.0) | Multidisciplinary | 151 (61.6) |
| | | | Post pictorial illustration – Post written description | −5.602 | < 0.001 | Cross-disciplinary | 122 (49.8) | Cross-disciplinary | 3 (1.2) | Cross-disciplinary | 5 (2.0) |
| | | | | | | Interdisciplinary | 0 (0.0) | Interdisciplinary | 30 (12.2) | Interdisciplinary | 62 (25.3) |
| | | | | | | Transdisciplinary | 0 (0.0) | Transdisciplinary | 6 (2.4) | Transdisciplinary | 21 (6.6) |

Several scholars asserted that awareness of the disciplinary collaboration models is a fundamental indicator of readiness to practice successfully in a healthcare team [19,20]. A considerable number of our study participants reported that they were aware of the collaborative practice models, especially, intradisciplinary, multidisciplinary, and interdisciplinary collaboration. However, another study among medical doctors in the obstetrics and gynecology unit of a tertiary hospital in Nigeria found inadequate levels of awareness of the disciplinary collaboration constructs among almost half of the respondents [8]. Perhaps, one's profession may influence their awareness level. For instance, we found that being a medical radiographer was predictive of lower awareness level. Building on our experience as clinicians working in Nigeria, we believe that this low awareness level among HCPs such as medical radiographers could be as a result of their job descriptions. Historically, medical radiographers practicing in Nigeria are often confined to their medical imaging rooms and do not participate actively in the general ward rounds. Consequently, they may not be aware of the disciplinary collaborations aside their interactions with radiologists, patients and other staff in the radiology department.

We also found that having at least a Bachelor's degree and exposure to transdisciplinary education predicted better awareness of the disciplinary collaboration constructs. Awareness building, knowledge and competencies needed to prepare practitioners for collaborative practices, such as transdisciplinary practice, are expected to be acquired through education [14]. A study among medical and engineering students found that students who participated in a transdisciplinary course demonstrated better awareness and attitude as well as readiness for high-performing collaborative practice models such as transdisciplinary [14]. These findings emphasize an imperative for healthcare training institutions in Nigeria to incorporate inter-/transdisciplinary education into their curriculum. We offer that this could be achieved by introducing a standalone course or integrating it into existing courses, accompanied by a practical component at the entry-level programs, to provide students with hands-on experience.

Our participants' perceived barriers to disciplinary collaboration in their practice setting include inadequate specialists or HCPs with geriatrics certification, lack of integrated EMRs, and inequitable remuneration formula. These findings corroborate previous studies on disciplinarity in healthcare [3,21]. Similarly, existing studies [3,13] also reported issues related to lack of pay parity as well as power imbalances between medical doctors and other healthcare professionals in Nigeria. Traditionally, these doctors are trained to belief they are the leaders of clinical teams. However, implementing a transdisciplinary practice model in this context may offset the power imbalances as transdisciplinarity blurs disciplinary boundaries, encourages horizontal power relation and situational leadership role, and allows any HCP to assume leadership within clinical teams.

Furthermore, some scholars argue that major organizational barriers such as resource constraints, organizational policies, and team structure posed a greater barrier compared to interpersonal characteristics [20]. Structural barriers could

impede collaborative practice, especially in a fragmented healthcare system [22] like in Nigeria. The issue of resource constraint is pervasive across parastatals in Nigeria. The Nigerian healthcare system experiences a spate of industrial actions due to issues related to inadequate and delayed payment of HCPs' and healthcare sector underfunding [23]. Targeted investments in training and absorbing geriatrics specialists and integrating EMRs into geriatric care settings seems far-fetched. Successive governments have not demonstrated sufficient political will to reform the healthcare system or improve investment in healthcare [23]. The plethora of issues affecting the healthcare systems constitutes tremendous roadblocks to achieving robust reforms. For instance, another study reported that, due to systemic challenges, HCPs and the hospital managements may not be motivated to pursue innovations to foster comprehensive and integrated care for older adults but rather maintain the traditional fragmented practice model [22].

We suggest a top-down solution to the aforementioned barriers since the prominent issues are within the purview of the government. Precisely, some of strategies should be related to the healthcare budget allocation, investment in healthcare infrastructure, and enacting educational and practice policies that support high-level collaborative practice, including promoting inter-/trans-disciplinary practice as Nigeria's standard practice for the care of older adults. Enabling environments (in training and practice) should be created for HCPs to acquire knowledge and skills pertaining to collaborative practice so as to facilitate cohesion, task sharing, shared goals, satisfaction with practice experience, and improved care outcomes [24].

## Strength and limitations

The present study is the first quantitative inquiry in Nigeria that studied the awareness of a broad range of HCPs and delineated the levels of disciplinary collaboration. For example, Iyoke and colleagues [8] did not consider the different levels of disciplinary collaboration but sampled the perspectives of only medical doctors. Also, another study in the South-Western region of Nigeria focused on medical doctors and nurses [13].

Our study was restricted to one healthcare facility. It may raise concerns about the generalizability of the study findings to the broader Nigerian context. However, our results corroborate the findings of previous studies on disciplinary collaboration in this clime [8,13]. The similarities suggest pervasive institutional factors may have shaped the healthcare delivery model and experiences in the country. Thus, this present study results may be extrapolated beyond the study site to other urban, tertiary hospitals. The results, on the other hand, may not apply to, or be representative of, certain settings such as primary and secondary healthcare facilities, as well as rural and geographically isolated and disadvantaged areas. We could not find any study on disciplinary collaboration in these settings. We recommend future studies to investigate the practice condition, team dynamics, and unique experiences in these areas.

We did not explore the participant's views regarding the facilitators of higher collaborative practice in the study site. Our decision was informed by previous study which revealed that Nigerian healthcare education and practice have not yet evolved into adopting high level collaborative practice models [3]. Also, our allusion above regarding overwhelming institutional constraints (e.g., lack of interprofessional education and policy and infrastructure to support collaborative practice) in the Nigerian healthcare ecosystem eclipses the need to examine the facilitators.

## Study implications

Our study revealed that high levels of disciplinary collaboration (e.g., transdisciplinary) are not currently practiced in the study setting as is the case in most Nigerian hospitals. We found that the capacity to adopt transdisciplinary practice is limited largely by systemic barriers. The persistence of systemic limitations to certain collaborative practices suggests that Nigerian healthcare policymakers may not yet recognize the need to adopt these practice models, especially, the benefits they offer, such as improving the overall effectiveness of healthcare services and patient satisfaction. It is also possible that some policymakers see collaborative practices as sources of increased financial pressure due to an expanded workforce such as increased and the associated costs of remuneration, incentives, and overhead.

It is imperative for the Nigerian healthcare policymakers to learn from other countries that are ahead in the implementation of interprofessional team-based care models. Canada, Sweden, United Kingdom, Japan, etc., are making great strides in the implementation of integrated care models and continue to refine their care delivery models to enhance performance [25]. To ameliorate the Nigerian challenges to actualizing high-level collaborative practice, organized interests in the care of older adults domain may need to exert pressure on policymakers to introduce transdisciplinary education in health professions' education curricula, and reform the healthcare delivery and financial arrangements to support transdisciplinary collaboration.

## Conclusion

In conclusion, the study participants demonstrate a good awareness of intradisciplinary, multidisciplinary, and interdisciplinary; but not cross-disciplinary and transdisciplinary collaboration. Multidisciplinary collaboration seemed to be common in the study setting. The lack of supportive structures and levers to encourage high-performing disciplinary practices (interdisciplinary and transdisciplinary) constitute enormous impediments to advancements in healthcare delivery in the study site but also in other tertiary hospitals across the country. Specific systemic and organizational issues identified in this study indicate key intervention areas for healthcare investment and policies in Nigeria.

## Supporting information

**S1 File. Questionnaire development: face and content validity.**
(PDF)

## Acknowledgments

We greatly appreciate Tasnia Noshin for proofreading the draft manuscript. We also appreaciate the staff and Management of Nnamdi Azikiwe University Teaching Hospital for their support during the data collection.

## Author contributions

**Conceptualization:** Augustine Chukwuebuka Okoh, Chukwunonso Geoffrey Okafor, Michael Kalu.

**Data curation:** Chukwunonso Geoffrey Okafor.

**Formal analysis:** Augustine Chukwuebuka Okoh, Chukwunonso Geoffrey Okafor, Michael Kalu.

**Methodology:** Augustine Chukwuebuka Okoh, Chukwunonso Geoffrey Okafor, Adaobi Odega, Stanley Maduagwu.

**Project administration:** Chukwunonso Geoffrey Okafor.

**Supervision:** Adaobi Odega, Stanley Maduagwu, Michael Kalu.

**Visualization:** Augustine Chukwuebuka Okoh, Chukwunonso Geoffrey Okafor.

**Writing – original draft:** Augustine Chukwuebuka Okoh, Chukwunonso Geoffrey Okafor.

**Writing – review & editing:** Augustine Chukwuebuka Okoh, Michael Kalu.

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
