## [Decision Letter · Decision Letter 0]

24 Mar 2025

PONE-D-24-38311Healthcare professionals’ perceptions of the disciplinary collaborations in older adults’ care during clinical practice in Nigeria: A cross-sectional studyPLOS ONE

Dear Dr. Okoh,

Thank you for submitting your manuscript to PLOS ONE. After careful consideration, we feel that it has merit but does not fully meet PLOS ONE’s publication criteria as it currently stands. Therefore, we invite you to submit a revised version of the manuscript that addresses the points raised during the review process.

**ACADEMIC EDITOR:** 

Overall assessment:

The manuscript addresses an important issue concerning interprofessional collaboration in geriatric healthcare in Nigeria. The study's focus on disciplinary collaboration levels and barriers provides valuable insights for strengthening geriatric healthcare delivery. However, revisions are required to improve clarity, methodological transparency, and consistency in reporting. Strengthening the discussion on policy implications and refining the statistical presentation will enhance the manuscript’s contribution to the field.

Major comments:

The manuscript discusses various levels of disciplinary collaboration (intradisciplinary, multidisciplinary, interdisciplinary, etc.), but the operational definitions and distinctions between these concepts need to be more explicitly clarified. Ensure that the terms are consistently applied throughout.The study employs a cross-sectional design, but further justification is needed regarding the sampling approach. Was convenience sampling the most appropriate method? How did the authors address potential biases in the participant selection process?The questionnaire development process should be elaborated. Was it pilot-tested or validated before use? If yes, provide details on how its reliability and validity were assessed.The results section presents descriptive and inferential statistics, but further explanation is needed regarding why specific tests (e.g., chi-square, logistic regression, Wilcoxon signed-rank test) were chosen for different analyses.In Table 3 (regression model), provide odds ratios with confidence intervals to enhance interpretability.The discussion should better connect the findings to practical implications for healthcare policy and practice in Nigeria. How do the barriers identified align with existing healthcare system challenges?While recommendations for training and policy change are mentioned, more concrete suggestions on how interdisciplinary collaboration can be integrated into healthcare education and practice would strengthen the discussion.

Minor comments:

The manuscript contains some grammatical errors and typographical issues (e.g., “transdiscplinary” should be “transdisciplinary” in the conclusion section of the abstract). A thorough proofreading is recommended.The abstract should explicitly state key numerical findings (e.g., awareness percentages) to provide a clearer summary of results.Ensure adherence to the PLOS ONE data availability policy—the statement should clarify how other researchers can access the dataset.

We look forward to receiving your revised manuscript.

Kind regards,

Gordon Dugle, Ph.D

Academic Editor

PLOS ONE

Journal Requirements:

3. Please include your tables as part of your main manuscript and remove the individual files. Please note that supplementary tables (should remain/ be uploaded) as separate "supporting information" files".

Reviewers' comments:

Reviewer's Responses to Questions

**Comments to the Author**

1. Is the manuscript technically sound, and do the data support the conclusions?

Reviewer #1: Yes

Reviewer #2: Yes

2. Has the statistical analysis been performed appropriately and rigorously? 

Reviewer #1: I Don't Know

Reviewer #2: Yes

3. Have the authors made all data underlying the findings in their manuscript fully available?

Reviewer #1: No

Reviewer #2: Yes

4. Is the manuscript presented in an intelligible fashion and written in standard English?

Reviewer #1: Yes

Reviewer #2: Yes

5. Review Comments to the Author

Reviewer #1: 1. Is the manuscript technically sound, and do the data support the conclusions?

The study "Healthcare professionals’ perceptions of the disciplinary collaborations in older adults’

care during clinical practice in Nigeria: A cross-sectional study" is relevant both in terms of the specific healthcare needs of older adults and in the broader context of improving healthcare systems in low-resource settings like Nigeria. It has the potential to contribute to knowledge, however, not all the data are provided.

2. Has the statistical analysis been performed appropriately and rigorously?

I don't know.

The discussion statistical analysis is good, but I cannot confirm because there are no tables. Also, they did not explain why they chose convenience sampling over probability sampling. They should explain that and justify their choice. Same applies to the sample units. Why did they settle on those they chose and not any other? These should be provided.

3. Have the authors made all data underlying the findings in their manuscript fully available?

No, the authors did not provide all available data. Even though there are indications of where tables and figures should be, no tables are found at the tail end of the document, nor are they found among the support documents. Only a figure is provided at the end of the document, and the only support document provided is the questionnaire.

4. Is the manuscript presented in an intelligible fashion and written in standard English?

Yes.

Largely the fashion and standard of English is good except for some few typos. E.g., these sentences 1. In paragraph 1 under discussion, the last two sentences "This highlighted the need to provide a concensus-contextual definitions of each concept to enhance collaborative that acualises the cultural distinctions across each practice settings" and

"Moreso, it is plausible that a detailed description of these disciplinary collaboration, placed in clinics is a starting point to enhancing HCPs’ understanding of these concepts" are difficult to understand. Same apples to the second sentence in paragraph 2. e.g., "A considerable number of our study participants reported that they were awareness of the collaborative practice models, especially, intradisciplinary, multidisciplinary, and interdisciplinary collaboration." does not read well.

These sentences should be rephrased.

Reviewer #2: The manuscript by Augustine Chukwuebuka Okoh and co-workers describes a cross-sectional survey among Nigerian Healthcare professionals to assess their perception of disciplinary collaborations in older adult care. The language of the manuscript is clear and concise with well-explained methodology allowing the reader to appreciate the study and findings. I find the objectives and study design well-aligned and statistical tools appropriately applied to report suitable findings. Although part of a larger study, nevertheless the findings here contribute to the paucity of information on disciplinary collaboration in geriatric practice in Nigeria, by adding to an understanding of the barriers and challenges to adopting this approach to the care of older adults.

6. PLOS authors have the option to publish the peer review history of their article (what does this mean?). If published, this will include your full peer review and any attached files.

Reviewer #1: No

Reviewer #2: No

---

## [Author Response · Author response to Decision Letter 1]

17 Apr 2025

ACADEMIC EDITOR:

Overall assessment:

The manuscript addresses an important issue concerning interprofessional collaboration in geriatric healthcare in Nigeria. The study's focus on disciplinary collaboration levels and barriers provides valuable insights for strengthening geriatric healthcare delivery. However, revisions are required to improve clarity, methodological transparency, and consistency in reporting. Strengthening the discussion on policy implications and refining the statistical presentation will enhance the manuscript’s contribution to the field.

Authors response: We appreciate your favourable review of our manuscript. We also appreciate the time and energy you have taken to provide us with constructive comments for improvement.

Major comments:

1. The manuscript discusses various levels of disciplinary collaboration (intradisciplinary, multidisciplinary, interdisciplinary, etc.), but the operational definitions and distinctions between these concepts need to be more explicitly clarified. Ensure that the terms are consistently applied throughout.

Authors response: Thank you for your comment. Figure 1 offers additional clarity, delineating increasing level of close interaction between healthcare professionals from different professions, especially from multidisciplinary to transdisciplinary collaboration. We have also ensured that these terms are consistently applied throughout the paper.

2. The study employs a cross-sectional design, but further justification is needed regarding the sampling approach. Was convenience sampling the most appropriate method? How did the authors address potential biases in the participant selection process?

Authors response: Thank you for highlighting the mistake. We actually used a criterion-based purposive sampling technique. Regarding how we managed potential biases:

- Primarily, we sent out emails to a few potential participants who email addresses we obtained through our professional networks. Self-selection bias and gatekeeper bias could skew the sample toward overrepresentation of participants with certain desirable attributes who may differ systematically from the target population. To control for this influence, we engaged in direct/in-person recruitment visits to the hospital and met several clinical staff (potential participants). We also circulated the study advert on the staff WhatsApp groups and information boards at the hospital premises.

- We were mindful of sampling homogeneity; hence, we ensured that we achieved a diverse sample comprising doctors, physiotherapists, nurses, etc.

- We controlled confirmation bias as per how we defined the criteria or avoided seeking ideal participant who may confirm our hypothesis. To address this, we ensured that our sample was not too narrow but diverse and defined based on the existing literature (For example – Okoh AE, Akinrolie O, Bell-Gam HI, Adandom I, Ibekaku MC, Kalu ME. Nigerian healthcare workers’ perception of transdisciplinary approach to older adults’ care: A qualitative case study. Int J Care Coord. 2020; 205343452095436. doi:10.1177/2053434520954362)

3. The questionnaire development process should be elaborated. Was it pilot-tested or validated before use? If yes, provide details on how its reliability and validity were assessed.

Authors response: We did not find an established, standardized questionnaire to adapt for this study. Thus, we developed one based on questions elicited from reviewing the literature on disciplinary collaboration in healthcare. We did not include it in the main manuscript because of word limits. We provided a detailed description of the face and content validation process in the supplementary file. We also referred readers to the supplementary file (see ‘Questionnaire structure’, page 10).

4. The results section presents descriptive and inferential statistics, but further explanation is needed regarding why specific tests (e.g., chi-square, logistic regression, Wilcoxon signed-rank test) were chosen for different analyses.

Authors response: We are unable to include every detail in the manuscript because of word limits. Justifications for the statistical tests, include:

- Chi-square test is suitable for determining if there was a significant association between categorical variables.

- Logistical regression to predict their awareness of inter/transdisciplinary because we had a binary dependent variable (awareness [yes or no]). Also, other assumptions were met including independence of observations, no multicollinearity, and our sample size was large enough with respect to the number of variables.

- Freidman test is the nonparametric alternative to repeated measure ANOVA. We used it because mainly because we had five groups (i.e., more than two groups), and we treated the levels of disciplinary collaboration as ordinal variables. We also had three measurements timepoints: baseline measure followed by two post measures after pictorial and written descriptions of the disciplinary collaboration constructs.

- Wilcoxon Wilcoxon signed-rank test was used for post hoc analysis to identify where the significant difference lied from the baseline to the post measures. We used Wilcoxon signed-rank test (the nonparametric alternative to the paired t-test) because of the ordinal variables.

5. In Table 3 (regression model), provide odds ratios with confidence intervals to enhance interpretability.

Authors response: We have provided odds ratios with confidence intervals as advised (see page 13)

6. The discussion should better connect the findings to practical implications for healthcare policy and practice in Nigeria. How do the barriers identified align with existing healthcare system challenges?

Authors response: We have revised the discussion, connected our findings to the literature and the broader Nigerian policy and practice context.

7. While recommendations for training and policy change are mentioned, more concrete suggestions on how interdisciplinary collaboration can be integrated into healthcare education and practice would strengthen the discussion.

Authors response: We have integrated this suggestion in the discussion. We highlighted transdisciplinary practice curriculum content and transdisciplinary geriatric care practice standards. We also proposed using improved government investments and targeted incentives for transdisciplinary practice to enhance uptake.

Minor comments:

1. The manuscript contains some grammatical errors and typographical issues (e.g., “transdiscplinary” should be “transdisciplinary” in the conclusion section of the abstract). A thorough proofreading is recommended.

Authors response: Thank you for highlighting these grammatical and typographical errors. We have corrected them many more throughout the paper.

2. The abstract should explicitly state key numerical findings (e.g., awareness percentages) to provide a clearer summary of results.

Authors response: We have provided the numbers for the results (see Abstract, page 3)

3. Ensure adherence to the PLOS ONE data availability policy—the statement should clarify how other researchers can access the dataset.

Authors response: We have indicated that the dataset is available upon request by contacting the correspond author.

Review Comments to the Author

Reviewer #1: 1. Is the manuscript technically sound, and do the data support the conclusions?

The study "Healthcare professionals’ perceptions of the disciplinary collaborations in older adults’ care during clinical practice in Nigeria: A cross-sectional study" is relevant both in terms of the specific healthcare needs of older adults and in the broader context of improving healthcare systems in low-resource settings like Nigeria. It has the potential to contribute to knowledge, however, not all the data are provided.

Authors response: Thank you for your positive comments. We have added the tables that were missing initially. The dataset generated and analysed during this study would not be publicly available because individuals’ privacy could be compromised, but it would be available from the corresponding author upon reasonable request, while ensuring ethical guidelines are adhered to.

2. Has the statistical analysis been performed appropriately and rigorously?

I don't know.

The discussion statistical analysis is good, but I cannot confirm because there are no tables. Also, they did not explain why they chose convenience sampling over probability sampling. They should explain that and justify their choice. Same applies to the sample units. Why did they settle on those they chose and not any other? These should be provided.

Authors response: We have provided the tables as recommended. We chose a nonprobability sampling technique over probability sampling technique because we did not have direct access to the full contact list of care providers in the study setting, including their emails and occupation. We had a few email addresses (less than 15% of the total sample) through our professional networks. We relied on some staff to circulate the study advert in their WhatsApp groups. We posted adverts on notice boards and also went on site visits for some in-person recruitment efforts. We used criterion sampling as we have already described above because we sought to elicit diverse and representative data that offers rich insight from different disciplines. We believe that collaborative practice would be better studied by seeking insights from a sample that comprises diverse healthcare professional groups.

3. Have the authors made all data underlying the findings in their manuscript fully available?

No, the authors did not provide all available data. Even though there are indications of where tables and figures should be, no tables are found at the tail end of the document, nor are they found among the support documents. Only a figure is provided at the end of the document, and the only support document provided is the questionnaire.

Authors response: We have provided the tables at the tail end of the manuscript.

4. Is the manuscript presented in an intelligible fashion and written in standard English?

Yes.

Largely the fashion and standard of English is good except for some few typos. E.g., these sentences 1. In paragraph 1 under discussion, the last two sentences "This highlighted the need to provide a concensus-contextual definitions of each concept to enhance collaborative that acualises the cultural distinctions across each practice settings" and "Moreso, it is plausible that a detailed description of these disciplinary collaboration, placed in clinics is a starting point to enhancing HCPs’ understanding of these concepts" are difficult to understand. Same apples to the second sentence in paragraph 2. e.g., "A considerable number of our study participants reported that they were awareness of the collaborative practice models, especially, intradisciplinary, multidisciplinary, and interdisciplinary collaboration." does not read well.

Authors response: Thank you for highlighting these grammatical and typographical errors. We have corrected them many more throughout the paper.

These sentences should be rephrased.

Reviewer #2: The manuscript by Augustine Chukwuebuka Okoh and co-workers describes a cross-sectional survey among Nigerian Healthcare professionals to assess their perception of disciplinary collaborations in older adult care. The language of the manuscript is clear and concise with well-explained methodology allowing the reader to appreciate the study and findings. I find the objectives and study design well-aligned and statistical tools appropriately applied to report suitable findings. Although part of a larger study, nevertheless the findings here contribute to the paucity of information on disciplinary collaboration in geriatric practice in Nigeria, by adding to an understanding of the barriers and challenges to adopting this approach to the care of older adults.

Authors response: Thank you for your favourable review of our manuscript.

---

## [Decision Letter · Decision Letter 1]

26 Aug 2025

PONE-D-24-38311R1Healthcare professionals’ perceptions of the disciplinary collaborations in older adults’ care during clinical practice in Nigeria: A cross-sectional studyPLOS ONE

Dear Dr. Okoh,

Thank you for submitting your manuscript to PLOS ONE. After careful consideration, we feel that it has merit but does not fully meet PLOS ONE’s publication criteria as it currently stands. Therefore, we invite you to submit a revised version of the manuscript that addresses the points raised during the review process.

Reviewer 1 has raised some issues and I resonate with those comments. I invite you to revise those comments. Please get your draft proofread from a native English speaker and duly acknowledge that person within your draft.

We look forward to receiving your revised manuscript.

Kind regards,

Shekhar Chauhan

Academic Editor

PLOS ONE

Journal Requirements:

Additional Editor Comments:

Dear Augustine,

Thank you for submitting the revision earlier. While reviewer showed positive response to your revised draft, he still has some concerns for which I again invite you to go through those comments and revise the draft accordingly.

Reviewers' comments:

Reviewer's Responses to Questions

**Comments to the Author**

1. If the authors have adequately addressed your comments raised in a previous round of review and you feel that this manuscript is now acceptable for publication, you may indicate that here to bypass the “Comments to the Author” section, enter your conflict of interest statement in the “Confidential to Editor” section, and submit your "Accept" recommendation.

Reviewer #1: (No Response)

Reviewer #3: All comments have been addressed

2. Is the manuscript technically sound, and do the data support the conclusions?

Reviewer #1: Yes

Reviewer #3: Yes

3. Has the statistical analysis been performed appropriately and rigorously? 

Reviewer #1: I Don't Know

Reviewer #3: Yes

4. Have the authors made all data underlying the findings in their manuscript fully available?

Reviewer #1: Yes

Reviewer #3: Yes

5. Is the manuscript presented in an intelligible fashion and written in standard English?

Reviewer #1: Yes

Reviewer #3: Yes

6. Review Comments to the Author

Reviewer #1: Overall assessment: The study topic “Healthcare professionals’ perceptions of the disciplinary collaborations in older adults’ care during clinical practice in Nigeria” is interesting and highly relevant. If shaped well, the study stands a chance of contributing significantly to knowledge in the healthcare sector, particularly in enhancing healthcare provision for older adults. The contents are presented with a reasonable level of structure and detail, except for some few issues that limit its rigor and clarity. Below are my comments:

1. The topic implies the study seeks to establish “Healthcare professionals’ perceptions of the disciplinary collaborations in older adults’care during clinical practice in Nigeria. Measurement of perception is different from measurement of awareness. I have a challenge understanding how perception can be taken as awareness.

2. The rationale for choosing a cross-sectional design is not clearly stated. Given the goals (e.g., assessing perceptions and testing influence of interventions like pictorial/written descriptions), a mixed-method or longitudinal design could potentially provide more depth so, if you used cross-sectional design, you need to justify the use of a cross-sectional design.

3. The calculation of the sample size is not clear. Though they used a formula for an unknown population, the study should clearly show how the calculation was done. For instance, the assumptions such as expected proportions are not stated.

4. There is a difference between convenience sampling and purposive sampling. With purposive sampling you choose those with key information. But the process described with flyers distributed for people who “ self-identified as licensed professional in their respective profession at the time of this study, as no records were publicly available to verify their licensure” and who are willing to participate, indicates it is convenience sampling. Purposive sampling chooses key informants that the researchers know and are aware of.

5. Some of the issues raised by reviewers have not been addressed. E.g.,

a. About pilot testing the questionnaire, this has also not been addressed.

b. Relating to the results section where further explanation is needed and explanation for specific tests such as chi-square, logistic regression, Wilcoxon signed-rank test) were chosen for different analyses.

These have also not been addressed with a lame explanation of word count limit.

6. Clarity of language: The paper needs to be proofread. For instance the opening statement of the introduction does not read well. E.g. “Globally, the population of older adults is increasing rapidly, by 2050, and the number of people aged 60 years and over is projected to outnumber children aged 0-14 [1].”

Reviewer #3: Thank you for inviting me to review this manuscript on Healthcare professionals’ perceptions of the disciplinary collaborations in older adults’ care during clinical practice in Nigeria: A cross-sectional study. While this work provides a significant information about the perception of healthcare professionals about collaboration to support older adults in Nigeria, it further amplifies the voices of healthcare professionals whose voices remain invisible in research.

7. PLOS authors have the option to publish the peer review history of their article (what does this mean?). If published, this will include your full peer review and any attached files.

Reviewer #1: No

Reviewer #3: No

---

## [Author Response · Author response to Decision Letter 2]

3 Nov 2025

Manuscript ID: PONE-D-24-38311R1

Response to Reviewers

Additional Editor Comments:

Thank you for submitting the revision earlier. While reviewer showed positive response to your revised draft, he still has some concerns for which I again invite you to go through those comments and revise the draft accordingly.

We appreciate your kind words and appreciate the time and energy you have taken to provide us with constructive feedback. We have made concerted efforts to address Reviewer #1’s concerns.

Please get your draft proofread from a native English speaker and duly acknowledge that person within your draft.

We have had a native speaker (who was born and has lived all their life in Canada) proofread the manuscript, and they have been acknowledged.

Review Comments to the Author

Reviewer #1: Overall assessment: The study topic “Healthcare professionals’ perceptions of the disciplinary collaborations in older adults’ care during clinical practice in Nigeria” is interesting and highly relevant. If shaped well, the study stands a chance of contributing significantly to knowledge in the healthcare sector, particularly in enhancing healthcare provision for older adults. The contents are presented with a reasonable level of structure and detail, except for some few issues that limit its rigor and clarity. Below are my comments:

1. The topic implies the study seeks to establish “Healthcare professionals’ perceptions of the disciplinary collaborations in older adults’care during clinical practice in Nigeria. Measurement of perception is different from measurement of awareness. I have a challenge understanding how perception can be taken as awareness.

Thank you for your comment. We actually examined participants’ awareness level. We have replaced mentions of ‘perception’ in the manuscript with ‘awareness’.

2. The rationale for choosing a cross-sectional design is not clearly stated. Given the goals (e.g., assessing perceptions and testing influence of interventions like pictorial/written descriptions), a mixed-method or longitudinal design could potentially provide more depth so, if you used cross-sectional design, you need to justify the use of a cross-sectional design.

Thank you for point this out. After due consideration, we believe that a multi-method study would be a more accurate description of the study approach. It comprises both cross-sectional and pretest-posttest study design, with the pre-post component testing the influence of the intervention.

3. The calculation of the sample size is not clear. Though they used a formula for an unknown population, the study should clearly show how the calculation was done. For instance, the assumptions such as expected proportions are not stated.

Thank you for your comment. We have added the formular and outline parameter as follows:

“The sample size (n) was calculated using the sample size determination formula for an unknown population as described by Naing et al.

n = Z2 P(1 – P)

d2

We assumed a Z statistic for a level of confidence (Z) of 95% (95% CI corresponds to Z value of 1.96), a degree of precision (d) of 5%, and expected prevalence or proportion (P) as 80%, thus, the study estimated sample size was 245.”

4. There is a difference between convenience sampling and purposive sampling. With purposive sampling you choose those with key information. But the process described with flyers distributed for people who “ self-identified as licensed professional in their respective profession at the time of this study, as no records were publicly available to verify their licensure” and who are willing to participate, indicates it is convenience sampling. Purposive sampling chooses key informants that the researchers know and are aware of.

Thank you for the correction. We have reframed the sampling technique as convenience sampling.

5. Some of the issues raised by reviewers have not been addressed. E.g.,

a. About pilot testing the questionnaire, this has also not been addressed.

While your question was about pilot testing, the questionnaire, we can only report what we did – face and content validity. A detailed description of the face and content validation process can be found in the supplementary file. We also referred readers to the supplementary file (see ‘Questionnaire structure’, page 10).

b. Relating to the results section where further explanation is needed and explanation for specific tests such as chi-square, logistic regression, Wilcoxon signed-rank test) were chosen for different analyses.

Justifications for the statistical tests, include:

- Chi-square test is suitable for determining if there was a significant association between categorical variables.

- Logistical regression to predict their awareness of inter/transdisciplinary because we had a binary dependent variable (awareness [yes or no]). Also, other assumptions were met including independence of observations, no multicollinearity, and our sample size was large enough considering the number of independent variables in the model (6 variables).

- Friedman test is the nonparametric alternative to repeated measure ANOVA. We used it because mainly because we had five groups (i.e., more than two groups), and we treated the levels of disciplinary collaboration as ordinal variables. We also had three measurements timepoints: baseline measure followed by two post measures after pictorial and written descriptions of the disciplinary collaboration constructs.

- Wilcoxon Wilcoxon signed-rank test was used for post hoc analysis to identify where the significant difference lied from the baseline to the post measures. We used Wilcoxon signed-rank test (the nonparametric alternative to the paired t-test) because of the ordinal variables.

These have also not been addressed with a lame explanation of word count limit.

The rationale for using each statistical test has been concisely described in the data analysis sub-section.

6. Clarity of language: The paper needs to be proofread. For instance the opening statement of the introduction does not read well. E.g. “Globally, the population of older adults is increasing rapidly, by 2050, and the number of people aged 60 years and over is projected to outnumber children aged 0-14 [1].”

Thank you for the comment. A native English language speaker has proofread the manuscript thoroughly.

Reviewer #3: Thank you for inviting me to review this manuscript on Healthcare professionals’ perceptions of the disciplinary collaborations in older adults’ care during clinical practice in Nigeria: A cross-sectional study. While this work provides a significant information about the perception of healthcare professionals about collaboration to support older adults in Nigeria, it further amplifies the voices of healthcare professionals whose voices remain invisible in research.

Thank you for your favourable review of our manuscript.

Reviewer #2: The manuscript by Augustine Chukwuebuka Okoh and co-workers describes a cross-sectional survey among Nigerian Healthcare professionals to assess their perception of disciplinary collaborations in older adult care. The language of the manuscript is clear and concise with well-explained methodology allowing the reader to appreciate the study and findings. I find the objectives and study design well-aligned and statistical tools appropriately applied to report suitable findings. Although part of a larger study, nevertheless the findings here contribute to the paucity of information on disciplinary collaboration in geriatric practice in Nigeria, by adding to an understanding of the barriers and challenges to adopting this approach to the care of older adults.

Thank you for your favourable review of our manuscript.

---

## [Decision Letter · Decision Letter 2]

4 Mar 2026

PONE-D-24-38311R2Healthcare professionals’ awareness of the levels of disciplinary collaboration in older adults’ care during clinical practice in Nigeria: A multi-method studyPLOS One

Dear Dr. Okoh,

Thank you for submitting your manuscript to PLOS ONE. After careful consideration, we feel that it has merit but does not fully meet PLOS ONE’s publication criteria as it currently stands. Therefore, we invite you to submit a revised version of the manuscript that addresses the points raised during the review process.

We look forward to receiving your revised manuscript.

Kind regards,

Confidence Alorse Atakro, Ph.D

Academic Editor

PLOS One

Journal Requirements:

Reviewers' comments:

Reviewer's Responses to Questions

**Comments to the Author**

1. If the authors have adequately addressed your comments raised in a previous round of review and you feel that this manuscript is now acceptable for publication, you may indicate that here to bypass the “Comments to the Author” section, enter your conflict of interest statement in the “Confidential to Editor” section, and submit your "Accept" recommendation.

Reviewer #3: All comments have been addressed

2. Is the manuscript technically sound, and do the data support the conclusions?

Reviewer #3: Yes

3. Has the statistical analysis been performed appropriately and rigorously? 

Reviewer #3: Yes

4. Have the authors made all data underlying the findings in their manuscript fully available?

Reviewer #3: Yes

5. Is the manuscript presented in an intelligible fashion and written in standard English?

Reviewer #3: Yes

6. Review Comments to the Author

Reviewer #3: I have gone through the revised version of the manuscript, and I have ascertained that the authors have done an amazing job. This manuscript will have significant contribution to wider readers in Africa.

7. PLOS authors have the option to publish the peer review history of their article (what does this mean?). If published, this will include your full peer review and any attached files.

Reviewer #3: **Yes:** Oluwagbemiga Oyinlola

---

## [Author Response · Author response to Decision Letter 3]

4 Mar 2026

Editor Comments:

… we invite you to submit a revised version of the manuscript that addresses the points raised during the review process.

Authors' response: We appreciate the time and energy you have dedicated to this manuscript’s review. We have cleaned the references, and no further changes were requested.

Journal Requirements:

Authors' response: No new references were suggested by the reviewer.

Authors' response: The reference list has been cleaned and duplicates remove.

Also, the following references are no longer available online:

• HelpAge. Country ageing data | Data | Global AgeWatch Index 2014. 2019 [cited 15 Aug 2020] pp. 1–4. Available: https://www.helpage.org/global-agewatch/population-ageing-data/countryageing-data/?country=South%2BAfrica

They have been replaced with up-to-date sources:

• Navaneetham K, Arunachalam D. Global Population Aging, 1950-2050. Handbook of Aging, Health and Public Policy: Perspectives from Asia. 2025; 99–116. doi:10.1007/978-981-99-7842-7_154

The following bibliographic details have been updated:

• United Nations Department of Economic and Social Affairs. World Population Ageing 2019. World population ageing 2019. 2019.

As seen below:

• United Nations Department of Economic and Social Affairs. World Population Ageing 2019. World population ageing 2019. 2019. Available: https://digitallibrary.un.org/record/3846855

Reviewers’ Comments:

1. If the authors have adequately addressed your comments raised in a previous round of review and you feel that this manuscript is now acceptable for publication, you may indicate that here to bypass the “Comments to the Author” section, enter your conflict of interest statement in the “Confidential to Editor” section, and submit your "Accept" recommendation.

Reviewer #3: All comments have been addressed

Authors' response: Thank you for your favourable review of our manuscript.

2. Is the manuscript technically sound, and do the data support the conclusions?

Reviewer #3: Yes

Authors' response: Thank you for your favourable review of our manuscript.

3. Has the statistical analysis been performed appropriately and rigorously?

Reviewer #3: Yes

Authors' response: Thank you for your favourable review of our manuscript.

4. Have the authors made all data underlying the findings in their manuscript fully available?

Reviewer #3: Yes

Authors' response: Thank you for your favourable review of our manuscript.

5. Is the manuscript presented in an intelligible fashion and written in standard English?

Reviewer #3: Yes

Authors' response: Thank you for your favourable review of our manuscript.

6. Review Comments to the Author

Reviewer #3: I have gone through the revised version of the manuscript, and I have ascertained that the authors have done an amazing job. This manuscript will have significant contribution to wider readers in Africa.

Authors' response: Thank you for your favourable review of our manuscript.

7. PLOS authors have the option to publish the peer review history of their article (what does this mean?). If published, this will include your full peer review and any attached files.

Do you want your identity to be public for this peer review? For information about this choice, including consent withdrawal, please see our Privacy Policy.

Reviewer #3: Yes: Oluwagbemiga Oyinlola

Authors' response: Thank you for your favourable review of our manuscript.

---

## [Editor Report · Decision Letter 3]

22 Apr 2026

Healthcare professionals’ awareness of the levels of disciplinary collaboration in older adults’ care during clinical practice in Nigeria: A multi-method study

PONE-D-24-38311R3

Dear Dr. Okoh,

We’re pleased to inform you that your manuscript has been judged scientifically suitable for publication and will be formally accepted for publication once it meets all outstanding technical requirements.

Kind regards,

Confidence Alorse Atakro, Ph.D

Academic Editor

PLOS One

Additional Editor Comments (optional):

Manuscript is ready for publication. Please format in line with the journal's guidelines.
---

## [Editor Report · Acceptance letter]

PONE-D-24-38311R3

PLOS One

Dear Dr. Okoh,

I'm pleased to inform you that your manuscript has been deemed suitable for publication in PLOS One. Congratulations! Your manuscript is now being handed over to our production team.

Kind regards,

on behalf of

Dr. Confidence Alorse Atakro

Academic Editor

PLOS One